# Unusual Square Pyramidal Chalcogenide Mo_5_ Cluster with Bridging Pyrazolate-Ligands

**DOI:** 10.3390/ijms24043440

**Published:** 2023-02-08

**Authors:** Iulia V. Savina, Anton A. Ivanov, Darya V. Evtushok, Yakov M. Gayfulin, Andrey Y. Komarovskikh, Mikhail M. Syrokvashin, Mariia N. Ivanova, Igor P. Asanov, Ilia V. Eltsov, Natalia V. Kuratieva, Yuri V. Mironov, Michael A. Shestopalov

**Affiliations:** 1Nikolaev Institute of Inorganic Chemistry of Siberian Branch of Russian Academy of Sciences, 3 Academician Lavrentiev Avenue, 630090 Novosibirsk, Russia; 2Department of Natural Sciences, Novosibirsk State University, 1 Pirogova st., 630090 Novosibirsk, Russia

**Keywords:** molybdenum, metal clusters, EPR, NMR, cyclic voltammetry, XPS, DFT calculations

## Abstract

The family of chalcogenide molybdenum clusters is well presented in the literature by a series of compounds of nuclearity ranging from binuclear to multinuclear articulating octahedral fragments. Clusters actively studied in the last decades were shown to be promising as components of superconducting, magnetic, and catalytic systems. Here, we report the synthesis and detailed characterization of new and unusual representatives of chalcogenide clusters: square pyramidal complexes [{Mo_5_(μ_3_-Se)^i^_4_(μ_4_-Se)^i^(μ-pz)^i^_4_}(pzH)^t^_5_]^1+/2+^ (pzH = pyrazole, i = inner, t = terminal). Individually obtained oxidized (2+) and reduced (1+) forms have very close geometry (proven by single-crystal X-ray diffraction analysis) and are able to reversibly transform into each other, which was confirmed by cyclic voltammetry. Comprehensive characterization of the complexes, both in solid and in solution, confirms the different charge state of molybdenum in clusters (XPS), magnetic properties (EPR), and so on. DFT calculations complement the diverse study of new complexes, expanding the chemistry of molybdenum chalcogenide clusters.

## 1. Introduction

Molybdenum chalcogenide clusters have been known since the 1960s and represent a functional class of cluster compounds [1,2]. Beginning with the well-known Chevrel phases [3], the chemistry of such clusters has evolved into a large variety of compounds with different structures and compositions. The consistent development of the chemical manipulation of these substances has led to a transition from polymeric insoluble compounds to molecular discrete clusters [2], which are often soluble in various solvents. The clusters containing {Mo_3_Q_7_}, {Mo_3_Q_4_} [4], and {Mo_6_Q_8_} (Q = O, S, Se, Te) [5,6,7] cores (Figure 1) are the most studied in this way. However, other chalcogenide cluster compounds with cores of {Mo_2_Q_4_} [8], {Mo_4_Q_4_} [4], {Mo_9_Q_11_} (Figure 1d) [9], {Mo_12_Q_14_} [10], and {Mo_15_Q_17_} [11] are also known in the literature. For many representatives of this class of compounds, reversible redox transformations [5,12,13,14] and switchable paramagnetism [15,16] are known, forming the pool of functional properties of clusters together with the superconducting properties of Chevrel phases [17]. In addition, chalcogenide Mo-clusters have already been shown to be promising components of catalytic systems for organic synthesis, hydrogen evolution reactions, and so on [18,19,20,21,22]. Various organic ligands are actively used to tune the properties of the obtained compounds [2], while complexes with ligands of the azole series are sparingly represented in the literature. The chemical diversity of azoles, the possibility of their modification, and their biological properties [23,24] result in coordination complexes based on them that are promising for the preparation of various functional compounds [25,26]. Moreover, the possibility of bridging coordination of azole ligands opens up the possibility of manifesting special effects. For example, the use of pyrazole as a bridging unit and magnetic coupler in dicopper(II) metallacyclic complexes has led to a broad class of compounds, the so-called dicopper(II) pyrazolenophanes, which are promising for the preparation of new functional magnetic materials [27].

Expanding the chemistry of molybdenum chalcogenide clusters, new and untypical representatives of such clusters were obtained in present work, namely the square pyramidal complexes [{Mo_5_(μ_3_-Se)^i^_4_(μ_4_-Se)^i^(μ-pz)^i^_4_}(pzH)^t^_5_]^1+/2+^ (denoted as **Mo_5_red** and **Mo_5_ox** for 1+ and 2+ complexes, respectively; pzH = pyrazole, i = inner, t = terminal). The **Mo_5_red** complex was synthesized from the octahedral cluster Mo_6_Br_12_ ([{Mo_6_Br^i^_8_}Br^t^_2_Br^t-t^_4/2_]) and characterized in detail, and its paramagnetic behavior and redox properties were investigated. Upon dissolution, the complex undergoes one-electron oxidation with a transition to the **Mo_5_ox** form, which has also been characterized in detail. Additionally, DFT calculations were performed for a detailed analysis of the geometry, electronic structures, and energy levels of the new compounds.

## 2. Results and Discussion

### 2.1. Synthesis and General Characterization

The method for obtaining Mo_5_ clusters was based on the synthetic approach conducted for Mo_6_ clusters [28]. At the first stage, the Mo_6_Br_12_ cluster interacts with in situ generated NaHSe in water [29], which results in an amorphous black compound insoluble in most solvents, whose composition, according to the elemental analysis, can be described as “NaMo_6_Se_8_Br_4_” (denoted as **Mo_6_**). Moreover, according to XPS data (discussed below), the compound is a mixture of chalcohalide molybdenum cluster of unknown composition and molybdenum oxide (probably MoO_3_). Due to the amorphous nature and chemical inertness of the compound, it was not possible to obtain its structure or to provide accurate information regarding the retaining of the octahedral structure. However, involving this amorphous product in a reaction with a melt of triphenylphosphine, as previously published, results in a neutral octahedral complex [{Mo_6_Se^i^_8_}(PPh_3_)^t^_6_] [28], which indirectly confirms the octahedral structure of **Mo_6_**. We expected that replacing triphenylphosphine with pyrazole would also lead to the formation of an octahedral complex, similar to rhenium clusters [30].

In a reaction of **Mo_6_** with pyrazole in a sealed ampoule at 200 °C for 48 h (see Section 3 for details), two types of crystals were formed, red and dark green, and an amorphous black product of unknown composition was observed. According to single-crystal X-ray diffraction analysis (SCXRD), the red crystals are a known compound of composition [Mo^V^_6_Mo^VI^_2_O_18_(pzH)_6_(μ-pz)_6_]·2pzH [31], which is a hybrid polyoxometalate (POM) built from Mo^V^_2_O_4_ fragments linked by oxo- and pyrazolate-ligands, as well as Mo^VI^ (Figure 2b). On the other hand, the dark green crystals correspond to a new compound [{Mo^III^_5_(μ_3_-Se)^i^_4_(μ_4_-Se)^i^(μ-pz)^i^_4_}(pzH)^t^_5_]Br·4pzH (denoted as **Mo_5_red**, Figure 2a), which was separated manually from the reaction mixture.

The X-ray diffraction pattern of the crystals of **Mo_5_red**, according to the powder X-ray diffraction analysis (PXRD) data, is in a good agreement with the theoretical one from SCXRD data, proving phase purity (Appendix A). In order to simplify the isolation of the complex, we extracted cluster from melt by various organic solvents. **Mo_5_red** is soluble in most solvents (DMSO, dichloromethane (DCM), acetone, acetonitrile (ACN), ethanol, methanol, water, etc.); however, the greatest stability, without changing the color or precipitation (usually a brown compound of unknown composition), was observed in DCM, acetone, ACN, and alcohols. Moreover, upon dissolution, the complex transforms into the oxidized form [{Mo^III^_4_Mo^IV^(μ_3_-Se)^i^_4_(μ_4_-Se)^i^(μ-pz)^i^_4_}(pzH)^t^_5_]^2+^, presumably due to oxygen from the air. Such a transition also agrees with the absorption spectra of dissolved **Mo_5_red** crystals in ACN or DCM in time (Appendix A). Thus, successive dissolution of the reaction mixture in acetonitrile and DCM (see Section 3 for details) yielded a crystalline powder [{Mo_5_(μ_3_-Se)^i^_4_(μ_4_-Se)^i^(μ-pz)^i^_4_}(pzH)^t^_5_]Br_2_·2H_2_O (denoted as **Mo_5_ox**). Crystals suitable for SCXRD of the compound were obtained by slow evaporation of a solution of the complex in acetone. Thus, the main product of the reaction in pyrazole melt is the reduced form **Mo_5_red**. Attempts to separate the cluster from the reaction mixture in solution resulted in oxidation of the cluster and formation of the **Mo_5_ox** form.

### 2.2. Crystal Structure of Compounds

According to SCXRD, **Mo_5_red** is built from a square pyramid of molybdenum atoms with four face-capped μ_3_-Se, one base-capped μ_4_-Se, and four edge-capped μ-pz ligands, forming an atypical, constructed by both chalcogen and organic moieties, cluster core {Mo_5_(μ_3_-Se)^i^_4_(μ_4_-Se)^i^(μ-pz)^i^_4_}^+^. Each molybdenum atom is additionally coordinated by a terminal pyrazole ligand (four basal and one apical) through a nitrogen atom in the second position of pyrazole (Figure 2a). The arrangement of inner face-capped μ_3_-Se ligands does not differ from structurally similar octahedral clusters with a core of {Mo_6_(μ_3_-Se)^i^_8_}^0^, while μ_4_-Se replaces one vertex of the octahedron (Figure 3a). Bridging pyrazolate ligands play the role of absent μ_3_-Se ligands and are located at an angle of 45.0° to the base of the Mo_5_ pyramid, while the outer basal ligands are almost parallel to the base of pyramid (deviation by 7.9°) (Figure 3b). The apical pyrazole ligand is disordered over four positions (rotation around the Mo-N bond) in such a way that the nitrogen atom in the first position and the carbon atom in the third position become indistinguishable (Figure 3). The packing of cluster cations in compound **Mo_5_red** is realized by N-H···N hydrogen bonds (N···N distance of 2.829 Å) and C-H···π stacking interactions (C-H···center of pyrazole distance of 2.612 Å) between terminal pyrazole ligands and solvated pyrazole molecules (Appendix A) forming infinite layers (Appendix A). The same solvate pyrazole molecules participate in N-H···Br hydrogen bonds (N···Br distance of 3.568 Å), connecting neighboring layers to form a 3D structure (Appendix A).

According to SCXRD, **Mo_5_ox** is built from the same cluster complex but with a charge of 2+, two bromine anions, and two water molecules. The geometry of the cluster changes slightly with a negligible “squeeze” of the pyramid Mo_5_ (see Table 1). The apical pyrazole ligand is disordered over two positions due to a small deviation of the pzH ring relative to the Mo-N bond (Appendix A). The packing of cluster cations in compound **Mo_5_ox** is realized by weak N-H···Br and C-H···Br hydrogen bonds (N···Br and C···Br distances of 3.267–3.280 and 3.775 Å, correspondingly) forming infinite chains (Appendix A). The chains are interconnected into a 3D structure by weak N/C-H···Br hydrogen bonds (N/C···Br distance of 3.353/3.706 Å, Appendix A). Disordered water molecules are located in the free space between the cluster complexes. According to PXRD, the diffraction pattern of powder **Mo_5_ox** is in good agreement with the theoretical one from SCXRD data (Appendix A).

The closest representatives of molybdenum cluster compounds for structural comparison are octahedral chalcogenide clusters [Mo_6_(μ_3_-Q)^i^_8_L^t^_6_] (Q = S, Se; L = organic ligands) and five-nuclear square pyramidal halide clusters [Mo_5_(μ-X)^i^_8_X^t^_5_]^n–^ (X = Cl, Br; *n* = 1, 2, 3). In fact, new compounds Mo_5_ are like a superposition of such types. Despite the rather high similarity between the upper part of the Mo_5_ pyramid and the Mo_6_ octahedron, including close Mo^bs^-Mo^ap^ (bs = basal, ap = apical) and Mo-Q distances, the Mo_5_ base is noticeably wider (even compared to halide Mo_5_-clusters), which apparently adapts to the coordination of the bridging pyrazolate ligands (Table 1). On the other hand, the oxidation state of molybdenum in new clusters is higher and lies between Mo_6_Q_8_/Mo_5_X_8_ and Mo_3_Q_4_/Mo_4_Q_4_ clusters, which can also affect the elongation of Mo^bs^-Mo^bs^ bond (Table 1). This is where the similarity with previously published compounds ends and features not previously presented for such clusters appear: (i) square pyramidal structure with chalcogenide ligands, (ii) μ_4_-Se ligand coordinated to Mo_4_ square, and (iii) participation of organic ligands in the formation of cluster core. The only example of Mo cluster compounds with μ-pz ligand is [Mo_4_S_4_(HB(pz)_3_)_4_(μ-pz)] obtained from [Mo_4_S_4_]_aquo_ and scorpionate ligand KHB(pz)_3_, hydrolysis of which during the synthesis results in pyrazolate-ligand [32]. Coordination of μ_4_-Se ligand was also mentioned for the series of molybdenum selenides cluster A_x_Mo_9_Se_11_ [9] and A_x_Mo_15_Se_19_ [33,34], where Mo_9_Se_11_ clusters are built from two octahedra connected along one face and additionally connected by μ_4_-Se coordinated to the sharing Mo-Mo edge and two Mo vertex of octahedrons (Figure 1d).

**Table 1 ijms-24-03440-t001:** Selected interatomic and average distances (Å), oxidation state, and valence electrons per cluster (VEC) for **Mo_5_red** and **Mo_5_ox** in comparison with literature data.

Compound	Mo-Mo (Average), Å		Mo-Q (Average), Å		Mo-N (Average), Å			Formal Mo Oxidation State	VEC	Formal Electrons per Mo-Mo Bond	Ref
	Base	Side	μ_3_-Q	μ_4_-Q	pzH^bs^	pzH^ap^	μ-pz				
**Mo_5_red**	2.8398(2)	2.6823(3)	2.5572(2)	2.5329(3)	2.217(2)	2.239(4)	2.184(2)–2.194(2) (2.189)	5 × Mo: +3	15	1.875	This work
**Mo_5_ox**	2.865(1)	2.660(1)–2.674(1) (2.667)	2.499(1)–2.5712(8) (2.5326)	2.547(1)–2.550(1) (2.548)	2.214(7)–2.230(7) (2.222)	2.25(1)	2.164(8)–2.209(7) (2.179)	4 × Mo: +3 1 × Mo: +4	14	1.750	This work
(Bu_4_N)_2_[Mo_5_Cl_8_Cl_5_]	2.602	2.563	–	–	–	–	–	4 × Mo: +2 1 × Mo: +3	19	2.375	[35]
Ag_3_._6_Mo_9_Se_11_	2.633–2.748 (2.701)		2.554–2.642 (2.594)	2.575–2.666 (2.620)	–	–	–	–	–	–	[9]
[Mo_6_Se_8_(PEt_3_)_6_]	2.697–2.705 (2.702)		2.550–2.569 (2.560)	–	–	–	–	4 × Mo: +3 2 × Mo: +2	20	1.667	[5]
[Mo_6_Se_8_(Ph_2_PC_2_H_4_CO_2_H)_6_]	2.695–2.705 (2.700)		2.531–2.574 (2.557)	–	–	–	–				[7]
Mo_6_Br_12_	2.630–2.640 (2.635)		–	–	–	–	–	6 × Mo: +2	24	2	[36]
(PPh_4_)_2_[Mo_3_Se_4_(C_2_O_4_)_3_(H_2_O)_3_]	2.811–2.826 (2.819)		2.445–2.455 (2.450)	–	–	–	–	3 × Mo: +4	6	2	[37]
(Bu_4_N)_2_[Mo_3_Se_7_Br_6_]	2.806–2.820 (2.813)		2.466–2.473 (2.470)	–	–	–	–				[38]
K_7_[Mo_4_Se_4_(CN)_12_]	2.900–2.925 (2.910)		2.493–2.522 (2.507)	–	–	–	–	3 × Mo: +3 1 × Mo: +4	11	1.833	[39]
[Mo_4_S_4_(HB(pz)_3_)_4_(μ-pz)]	2.659–2.952 (2.856)		2.304–2.404 (2.353)	–	2.19–2.34 (2.24)		2.20				[32]

### 2.3. DFT Calculations

For a more detailed characterization of the new cluster compounds, quantum chemical calculations were carried out (see Section 3 for details). The {Mo_5_Se_5_(pz)_4_}^+^ cluster core in the **Mo_5_red** cluster have an idealized point group symmetry C_4v_. The presence of a pzH molecule in apical positions lowers the symmetry of the discrete cluster to C_1_. However, rotational disorder of the pzH ligand allowed crystallization of the compounds **Mo_5_red** and **Mo_5_ox** in tetragonal and orthorhombic space groups, respectively, with 4-fold or 2-fold crystallographic axes passing through apical Mo and μ_4_-Se atoms. Optimization of the interatomic distances of the **Mo_5_red** cluster in C_1_ symmetry showed that the {Mo_5_} center contains four longer covalent bonds between Mo atoms in the basal plane (2.867(5) Å) and four shorter Mo_bs_–Mo_ap_ distances, having an average value of 2.668(1) Å (Appendix A). The calculated bond lengths are in good agreement with the crystallographic values (Table 1 and Appendix A). Taking into account that the typical length of a two-electron covalent bond between Mo atoms is close to 2.67–2.68 Å [5,6,40], we can assume that Mo^bs^–Mo^ap^ bonds have a bond order close to 1, while for the longer Mo^bs^–Mo^ap^ bonds, the bond order is 0.75.

One-electron oxidation of the cluster forming [{Mo_5_(μ_3_-Se)_4_(μ_4_-Se)(μ-pz)_4_}(pzH)_5_]^2+^ cation (**Mo_5_ox**) causes elongation of Mo^bs^–Mo^bs^ distances and shortening of Mo^bs^–Mo^ap^ ones; both effects are, however, quite small. The spread of bond lengths around the average values is negligible, indicating that the real symmetry of the {Mo_5_Se_5_(pz)_4_} core is close to the idealized C_4v_ one.

The electronic structure of the [{Mo_5_(μ_3_-Se)_4_(μ_4_-Se)(μ-pz)_4_}(pzH)_5_]^2+^ cluster cation displays a set of frontier orbitals that are primarily composed of d-orbitals of Mo atoms with a significant contribution of atomic orbitals of the inner ligands (Appendix A). A characteristic feature of the electronic structure is the presence of closely spaced LUMO and LUMO+1, separated from HOMO and LUMO+2 by wider energy gaps (Figure 4a). The corresponding values are 2.039, 0.544, and 1.378 eV for the HOMO–LUMO, LUMO–LUMO+1, and LUMO+1–LUMO+2 gaps, respectively. HOMO and HOMO–1 are bonding relative to the Mo–Mo and Mo–Se interactions, as well as Mo–N^μ-pz^ interactions. The μ_4_-Se ligand forms two-center bonds with each of the Mo atoms in the basal plane, while the μ_3_-Se ligands tend to form three-center Se–Mo–Se interactions. LUMO (HOMO for the **Mo_5_red** cluster) displays a bonding character between Mo^bs^ atoms, as well as along Mo–(μ_4_-Se) interactions, which agrees well with overall basal metal–ligand bond shortening in the experimental molecular structures upon reduction. LUMO+1 and LUMO+2 are strongly antibonding (Figure 4b).

The core of the new pentanuclear clusters has a distinguished apical direction in idealized symmetry: a rotational 4-fold axis, where the Mo^ap^ and μ_4_-Se atoms are located. We calculated the charges on atoms of the cluster core with use of the Bader method and analyzed the bond electron density by the electron localization function (ELF). Atomic charge analysis (Appendix A) showed that the Mo^ap^ and μ_4_-Se atoms have a significantly lower charge modulus compared to other Mo and Se atoms, respectively. Maps of ELF function demonstrate that the μ_4_-Se atom forms four equivalent two-centered interactions with Mo^bs^ atoms as donors (Appendix A). One can see that maximum localization basins V(Se,Mo) are displaced from the direct lines between Mo and Se atoms. This may indicate the formation of tense (banana) bonds.

The bond order analysis was performed for the cations [{Mo_5_(μ_3_-Se)_4_(μ_4_-Se)(μ-pz)_4_}(pzH)_5_]^+^ from **Mo_5_red** and [{Mo_5_(μ_3_-Se)_4_(μ_4_-Se)(μ-pz)_4_}(pzH)_5_]^2+^ from **Mo_5_ox** to elucidate the oxidation states of the constituent Mo ions (Table 2). Bond orders are shown separately for metal–metal and metal–ligand interactions of apical and basal Mo atoms, as well as total values. Although the cluster symmetry becomes lower with the one-electron oxidation process, this results in no observable difference in bond orders for the four Mo^bs^ atoms. The total bond orders for both species are close to the number of their valence electrons, while the valence violation for Mo-Mo and Mo-L bonds is clearly observed. This effect is known to be steric/electrostatic in nature, being the result of the expansion of the transition metal clusters due to the large size of surrounding anions and corresponding compression of the metal–ligand bonds [41,42]. The oxidation states are not easy to deduce for Mo in both cluster ions. A deviation from equal formal valence value +3 for all Mo atoms is observed.

### 2.4. X-ray Photoelectron Spectroscopy

The chemical states of the elements in the studied compounds were evaluated using X-ray photoelectron spectroscopy (XPS) (Table 3 and Appendix A, Figure 5 and Appendix A). The survey XPS spectra suggest the existence of Mo, Se, Br, C, O, and Na in the initial **Mo_6_** (Appendix A). The Mo3d region can be represented as a superposition of two groups of Mo3d_5/2–3/2_-lines in approximately equal ratio arising from nonequivalent molybdenum atoms with binding energy (BE) of 229.2–232.4 and 232.8–236.0 eV, respectively (Figure 5b). The first one can be assigned to molybdenum with both the +2 and +3 oxidation states according to literature (Table 3), which strongly depends on the ligand environment (halide or chalcogenide). The second doublet is assigned to Mo^6+^, probably in MoO_3_ form (the presence of a signal O 1s), which indicates the hydrolysis and destruction of part of the Mo_6_Br_12_ during the reaction in water. The presence of this component can also explain the low yield of the new pentanuclear complex (about 30%) and that of previously reported [{Mo_6_Se^i^_8_}(PPh_3_)^t^_6_] (about 33%). The Br3d region exhibits a superposition of two main doublets, Br3d_5/2–3/2_ (69.9–70.9 eV) and Br3d_5/2–3/2_ (68.2–69.3 eV), indicating the absence of inner Br-ligands (70.7–71.8 eV), in comparison with those for Mo_6_Br_12_ (Appendix A); this can be attributed to apically coordinated Br-ligands. Analysis of the Se3d region does not give an unambiguous answer about the coordination of the Se-ligand: doublet Se3d_5/2–3/2_ (54.1–55.0 eV) can be assigned to μ_3_-Se ligands, and doublet Se3d_5/2–3/2_ (55.3–56.1 eV) to Se_2_^2–^ or bridging Se-ligands.

The analysis of the XPS spectra of the new pentanuclear clusters is more informative (Figure 5 and Appendix A). Thus, the Mo3d region of **Mo_5_red** exhibited one main Mo3d_5/2–3/2_ (228.9–232.1 eV) doublet, while the Mo3d region of **Mo_5_ox** represented two groups of doublets Mo3d_5/2–3/2_ with a ratio of 4:1 and BE of 229.3–232.4 and 230.7–233.9 eV, respectively (Figure 5c,d). The first group of Mo3d-lines in **Mo_5_red** and **Mo_5_ox** is attributed to Mo^3+^, while the second one is only in **Mo_5_ox** and corresponds to Mo^4+^, thereby confirming the oxidation of the cluster core. The obtained binding energies are in good agreement with the previously reported data on the chalcogenide cluster compounds (Table 3). In addition, the Mo3d-region of both clusters contains low-intensity peaks (233.3–236.4 and 232.1–235.2 eV for **Mo_5_red** and **Mo_5_ox**, respectively), referred to Mo^6+^. This fact indicates the inevitable surface oxidation due to the contact with the air during the sample manipulation. The Se3d region (Appendix A) contains two groups of Se3d_5/2–3/2_ doublet (53.8–54.7/54.7–55.6 eV and 54.1–55.0/54.9–55.8 for **Mo_5_red** and **Mo_5_ox**, respectively) in the approximate ratio of 4:1, arising from μ_3_-Se and μ_4_-Se ligands, respectively. The slight differences observed in the Se3d-lines of **Mo_5_red** and **Mo_5_ox** indicate that the cluster core oxidation does not significantly affect the selenium charge state. Thus, the results obtained have additionally confirmed the charge and coordination state of the elements in the compounds studied.

### 2.5. NMR and HR-ESI-MS Spectroscopy

To confirm the composition and structure of the compounds, as well as to check the stability of the compounds in the solution, a detailed study of the cluster was carried out using multinuclear NMR-spectroscopy (^1^H, ^13^C, ^77^Se, Figure 6 and Appendix A) and high-resolution electrospray mass spectrometry (HR-ESI-MS) (Figure 7). Since **Mo_5_red** is paramagnetic (as will be discussed later), it has not been studied by NMR.

The ^1^H NMR spectrum of **Mo_5_ox** (Figure 6a) contains eight signals corresponding to protons of three different types of pyrazole ligands: basal and apical pyrazoles and bridging pyrazolate. The position of the signals of bridging ligands (6.26 and 7.53 ppm) almost does not differ from the position of the signals of free pyrazole (6.33 and 7.61 ppm), while signals from other ligands are more influenced by coordination to molybdenum. In the case of an apical ligand, the proton signals are shifted upfield (from 6.33 to 5.79 for H4 and from 7.61 to 6.36 and 7.10 for H3 and H5, respectively), while an opposite situation was observed for basal ligands, downfield shifting (from 6.33 to 6.85 for H4 and from 7.61 to 8.67 and 8.26 for H3 and H5, correspondingly), which indicates the different type of chemical environment of the ligands and their different reactivity. Indeed, during long-term storage of a solution of the cluster complex in methanol, we found the appearance of new signals slightly downfield-shifted in comparison with the initial one (Appendix A). Moreover, new signals were found only for μ-pz ligands and pzH^bs^, and release of free pzH was observed, which indicates lability of the pzH^ap^ ligand, which was presumably replaced by a solvent molecule or bromine.

In addition, the complex was characterized by ^77^Se spectroscopy (Figure 6b). In the wide range of the spectrum, only two signals were found at 1022 and 1880 ppm, which refer to two types of inner ligands, μ_4_-Se and μ_3_-Se, respectively. This method is rarely used in the chemistry of molybdenum cluster compounds. For example, for triangular clusters having Mo^IV^, the position of the signal of μ_3_-Se in ^77^Se NMR was found at 1356, 700, or 666 ppm for complexes [Mo_3_(μ_3_-Se)(μ-O)_3_(acac)_3_(py)_3_]PF_6_ [48], (Bu_4_N)_2_[Mo_3_(μ_3_-Se)(μ-Se_2_)_3_Br_6_] [38], and K_2_[Mo_3_(μ_3_-Se)(μ-Se_2_)_3_(C_2_O_4_)_3_] [37], respectively. The closest example (according to the charge state of the metal) is [{W^II^_2_W^III^_4_(μ_3_-Se)^i^_8_}(Ph_2_PC_2_H_4_COOH)^t^_6_] [49], for which the signal of μ_3_-Se was at 1018 ppm.

The main signal in the HR-ESI-MS spectrum (Figure 7 and Appendix A) of a solution of **Mo_5_ox** in acetonitrile belongs to the {[{Mo_5_Se_5_(pz)_4_}(pzH)_5_]Br}^+^ form, while other forms of much lower intensity can be attributed to one with solvate molecules of acetonitrile, without one terminal pzH-ligand, or to reduced form, all of which occur during sample ionization.

### 2.6. Magnetic Properties and EPR Spectroscopy

The magnetic properties of **Mo_5_red** were studied by magnetic susceptibility measurements and EPR spectroscopy. The temperature dependencies of the magnetic susceptibility were measured in two cycles: cooling (300–80 K) and heating (80–300 K). Then, the obtained values were averaged (Figure 8a). In the investigated temperature range of 80–300 K, the magnetic susceptibility obeys the Curie–Weiss law χ(T) = C·(T–Θ)^–1^. The approximation parameters (C and Θ) are shown in Figure 8a. The negative sign of the paramagnetic Curie temperature Θ = –6(4)K indicates antiferromagnetic ordering at low temperatures. The value of the effective moment is 1.78(2) μ_B_, which corresponds to 1.04(2) unpaired electrons per cluster. Using the isotropic value of *g*-tensor g_iso_ = 2.13, obtained by the EPR method (see below), the number of unpaired electrons per cluster is s = 0.94(2), which is in a good agreement with theoretical predictions.

At 300 K, the EPR spectrum of the **Mo_5_red** polycrystalline sample shows broadened lines (Appendix A). At 77 K, the lines narrow, and the observed signal (Figure 8b) clearly corresponds to a species with S = 1/2 and apically symmetric g-tensor (*g*_zz_ < *g*_xx_ = *g*_yy_). Using the least-squares fitting procedure implemented in the EasySpin program package, the spectrum was fitted with conventional spin Hamiltonian:(1)$$\hat{H}=\beta \hat{S}gH$$
where *β* is a Bohr magneton, *g* is a g-tensor, and *H* is a magnetic field. The obtained principal values of the *g*-tensor are *g*_xx_ = *g*_yy_ = 2.20 and *g*_zz_ = 1.99 (*g*_iso_ = 2.13). The estimated concentration of paramagnetic species is 93% of that theoretically predicted from the weight of the sample portion and molar mass, which is in a good agreement with the theoretical one considering the error of quantitative EPR spectroscopy.

The information about square pyramidal Mo_5_ clusters is scarce, and only a few examples are known in the literature. For the (Bu_4_N)_2_[Mo_5_Cl_8_Cl_5_] compound, the EPR spectra were reported, revealing a reverse situation of *g*_zz_ > *g*_xx_ = *g*_yy_ [35,50]. Another shape of the EPR spectrum can be explained by the different coordination of metal ions in the cluster structure. We performed DFT calculations for [{Mo_5_(μ_3_-Se)_4_(μ_4_-Se)(μ-pz)_4_}(pzH)_5_]^+^ cation to confirm the signal interpretation. The computation shows that an unpaired electron is localized mostly on the base of the Mo_5_ pyramid, occupying a d_xy_ orbital for each Mo located in the pyramid, with an axis directed to N of basal pyrazole ligand (Appendix A). For [{Mo_5_(μ_3_-Se)_4_(μ_4_-Se)(μ-pz)_4_}(pzH)_5_]^+^_,_ the calculated *g*-tensor eigenvalues are *g*_xx_ = 2.22, *g*_yy_ = 2.21, and *g*_zz_ = 1.96 (*g*_iso_ = 2.13), matching well to experimentally determined ones. The z-axis of the calculated *g*-tensor, which is almost apically symmetric, deviates from the structural 4-fold symmetry axis at around 1°.

### 2.7. Redox Properties

As noted earlier, molybdenum chalcogenide clusters exhibit reversible redox properties. Moreover, upon dissolution of **Mo_5_red**, the cluster core was oxidized by atmospheric oxygen. Therefore, redox properties of pentanuclear cluster in acetonitrile were studied by cyclic voltammetry (CV) (Figure 9 and Appendix A, Table 4). According to CV curves, **Mo_5_red** demonstrates quasi-reversible reduction (E_1/2_ = –0.57 V, transition from 15 to 16 VEC), and reversible oxidation (E_1/2_ = 0.11 V vs. Ag/AgCl, transition from 15 to 14 VEC to form [{Mo_5_Se_5_pz_4_}(pzH)_5_]^2+^) (Figure 9), followed by two irreversible oxidations (E_a_ = 0.77 V and E_a_ = 1.04 V) (Appendix A) corresponding to oxidation of Br^–^ and Br_3_^–^ to Br_2_ [51]. Undoubtedly, the CV curves of **Mo_5_red** and **Mo_5_ox** are the same, which confirms the assignment of 14 VEC form to **Mo_5_ox**, the reversibility of the process, and the preservation of the geometry of clusters. In addition, the low potential of oxidation of **Mo_5_red** supports the ability of the compound to be oxidized by molecular oxygen. Another form (16 VEC) can be attributed to cluster [{Mo_5_Se_5_pz_4_}(pzH)_5_]^0^, which was not yet isolated as an individual compound.

The formation of the neutral form (16 VEC) is also confirmed by the quasi-reversibility of the reduction of **Mo_5_red**, where the current of the reduction signal (–0.6 V) is more intense than the reverse oxidation (–0.54 V), which is apparently associated with the low solubility of the neutral form and its precipitation on the electrode surface. Despite numerous examples of studies of the redox properties of chalcogenide molybdenum clusters (Table 4), it is rather difficult to compare these results with that obtained for the new pentanuclear clusters, since the transitions usually strongly depend on the molybdenum charge state, cluster nuclearity, and ligand environment. However, some patterns can be found: [{Mo^III^_5_Se_5_pz_4_}(pzH)_5_]^+^ demonstrates one-electron reduction and oxidation similar to those reported for [Mo^III^_3_Mo^IV^Se_4_(edta)_2_]^3–^ and [Mo^II^_4_Mo^III^Br_13_]^2–^, but both signals are shifted to lower potentials (Δ ∽ 0.2–0.3 V).

## 3. Materials and Methods

### 3.1. Chemicals and Materials

Mo_6_Br_12_ was obtained by a reaction of metallic molybdenum with bromine [53]. All other reactants and solvents were purchased from Fisher (Hampton, NH, USA), Alfa Aesar (Haverhill, MA, USA), and Sigma-Aldrich (St. Louis, MO, USA) and used as received.

### 3.2. Syntheses

#### 3.2.1. “NaMo_6_Se_8_Br_4_” (Denoted as **Mo_6_**)

This compound was obtained according to the literature procedure, using NaBH_4_ instead of KBH_4_ [28]. Briefly, Se (300 mg, 3.80 mmol) and NaBH_4_ (245 mg, 6.48 mmol) were dissolved in degassed water (30 mL) in a flow of argon under stirring. Mo_6_Br_12_ (500 mg, 0.33 mmol) was added to the solution, and the reaction mixture was boiled for 2 h, cooled to room temperature, and allowed to stay overnight. The black amorphous product was centrifugated, washed with water, ethanol, and diethyl ether and dried in air. The product is insoluble in water and major organic solvents. Yield: 500 mg. EDS: Mo:Se:Br:Na atomic ratio was equal to 6:8.2:4.4:0.8. ICP-AES Calc. (found): Mo/Na 6.7 ± 0.5, Se/Mo 1.35 ± 0.02.

#### 3.2.2. [{Mo_5_(μ_3_-Se)^i^_4_(μ_4_-Se)^i^(μ-pz)^i^_4_}(pzH)^t^_5_]Br 4pzH (Denoted as **Mo_5_red**)

**Mo_6_** (50 mg) and pyrazole (150 mg) were heated in a sealed glass tube under ambient conditions at 200 °C for 2 days. The reaction mixture was slowly cooled to room temperature at a rate of 7.5 °C/h and washed with diethyl ether. Dark green crystals of **Mo_5_red** suitable for X-ray structural analyses were separated manually from black powder and byproduct’s red crystals. Yield: 20 mg (37% based on Mo_6_Br_12_). Anal. Calcd. for C_39_H_48_BrMo_5_N_26_Se_5_: C, 25.5; H, 2.6; N, 19.8. Found: C, 25.5; H, 2.8; N, 19.8. EDS: Mo:Se:Br atomic ratio was equal to 5:4.8:1.2. FTIR (KBr, cm^−1^): all expected peaks for the pyrazole ligand were observed (Appendix A). The TGA analysis indicates stability up to 100 °C, following by release of solvated pzH up to ∽ 180 °C, and decomposition of cluster (Appendix A). UV–vis (CH_3_CN): λ_max_, nm (ε, M^−1^ cm^−1^); 430 (4.6 × 10^3^) (Appendix A). The complex can also be obtained by a reaction of 200 mg **Mo_6_** with 200 mg pzH under the same conditions according to PXRD of the reaction mixture (Appendix A). However, in this case, it is difficult to separate the crystals from the reaction mixture, and when dissolved in organic solvents, the complex oxidizes to **Mo_5_ox** (see below).

#### 3.2.3. [{Mo_5_(μ_3_-Se)^i^_4_(μ_4_-Se)^i^(μ-pz)^i^_4_}(pzH)^t^_5_]Br_2_ 2H_2_O (Denoted as **Mo_5_ox**)

**Mo_6_** (200 mg) and pyrazole (200 mg) were heated in a sealed glass tube under ambient conditions at 200 °C for 2 days. The reaction mixture was slowly cooled to room temperature at a rate of 7.5 °C/h. As noted above, the complex **Mo_5_red** is formed at this stage. The reaction mixture was washed with diethyl ether and dissolved in 75 mL of acetonitrile, which is accompanied by oxidation of the complex with atmospheric oxygen. The solution was filtered off and evaporated until dry. Powder was dissolved in 50 mL of dichloromethane, filtered off, and evaporated until dry. The desired green product was washed with diethyl ether. The compound can be additionally purified from brown byproducts by column chromatography (eluent: mixture of DCM and ethanol 10:1). Yield: 66 mg (30% based on Mo_6_Br_12_). Anal. Calcd. for C_27_H_36_Br_2_Mo_5_N_18_Se_5_: C, 19.3; H, 2.2; N, 15.0. Found: C, 19.6; H, 2.3; N, 14.9. EDS: Mo:Se:Br atomic ratio was equal to 5:4.7:2.2. FTIR (KBr, cm^−1^): all expected peaks for the pyrazole ligand were observed (Appendix A). The TGA analysis revealed a weight loss of ∼1.9% from 25 to 120 °C (the calculated weight loss of 2 H_2_O is 2.1%) and stability of the complex up to 195 °C (Appendix A). ^1^H NMR (500 MHz, CD_3_OD) δ 5.79 (t, 1H, J = 2.41 Hz, H4-pzH^ap^), 6.26 (t, 4H, J = 2.06 Hz, H4-pz), 6.36 (d, 1H, J = 2.13 Hz, H3-pzH^ap^), 6.85 (t, 4H, J = 2.35 Hz, H4-pzH^bs^), 7.10 (d, 1H, J = 2.35 Hz, H5-pzH^ap^), 7.53 (d, 8H, J = 2.01 Hz, H3-, H5-pz), 8.26 (d, 4H, J = 2.35 Hz, H5-pzH^bs^), 8.67 (d, 4H, J = 1.84 Hz, H3-pzH^bs^). ^13^C NMR (126 MHz, CD_3_OD) δ 107.12 (C4-pzH^ap^), 108.38 (C4-pzH^bs^), 110.06 (C4-pz), 133.11 (C5-pzH^ap^), 133.96 (C5-pzH^bs^), 140.83 (C3-, C5-pz), 148.62 (C3-pzH^ap^), 149.11 (C3-pzH^bs^). ^77^Se NMR (95 MHz, SeO_2_) δ 1022 (μ_4_-Se), 1880 (μ_3_-Se). ^15^N NMR (51 MHz, HCONH_2_) δ 214 (NH-pzH^bs^), 216 (NH-pzH^ap^), 236 (N-pzH^bs^), 256 (N-pz). HR-ESI-MS (+) acetonitrile: 1563.3398 ({Mo_5_Se_5_(pz)_4_(pzH)_5_Br}^1+^), 1483.4160 ({Mo_5_Se_5_(pz)_4_(pzH)_5_}^1+^), 1536.3234 ({Mo_5_Se_5_(pz)_4_(pzH)_4_(CH_3_CN)Br}^1+^), 1415.3801 ({Mo_5_Se_5_(pz)_4_(pzH)_4_}^1+^), 1494.2979 ({Mo_5_Se_5_(pz)_4_(pzH)_4_Br}^1+^), 1456.4056 ({Mo_5_Se_5_(pz)_4_(pzH)_4_(CH_3_CN)}^1+^), 707.6890 ({Mo_5_Se_5_(pz)_4_(pzH)_4_}^2+^), 741.7084 ({Mo_5_Se_5_(pz)_4_(pzH)_5_}^2+^), 716.6967 ({Mo_5_Se_5_(pz)_4_(pzH)_4_(H_2_O)}^2+^), 673.6717 ({Mo_5_Se_5_(pz)_4_(pzH)_3_}^2+^), 694.1852 ({Mo_5_Se_5_(pz)_4_(pzH)_3_(CH_3_CN)}^2+^) (Appendix A). UV–vis (CH_3_CN): λ_max_, nm (ε, M^−1^ cm^−1^); 459 (5.1 × 10^3^) (Appendix A). The single crystals of **Mo_5_ox** suitable for X-ray structural analyses were obtained by slow evaporation of solution of cluster in acetone.

### 3.3. Physical Methods

Elemental analyses were obtained using a EuroVector EA3000 Elemental Analyser (S.p.A.,Milan, Italy). FTIR spectra were recorded on a Scimitar FTS 2000 (Digilab LLC, Canton, MA, USA). Energy-dispersive X-ray spectroscopy (EDS) was performed on a Hitachi TM3000 TableTop SEM (Hitachi High-Technologies Corporation, Tokyo, Japan) with Bruker QUANTAX 70 EDS equipment. Inductively coupled plasma atomic emission spectroscopy (ICP-AES) was carried out on a Thermo Scientific iCAP-6500 (Thermo Scientific, Waltham, MA, USA) high-resolution spectrometer with a cyclone-type spray chamber and a “SeaSpray” nebulizer. The spectra were obtained by axial plasma viewing. Deionized water (R ≈ 18 MΩ) was used to prepare the sample solutions. Absorption spectra were recorded on a Cary 60 UV–Vis Spectrophotometer (Agilent Technologies, Santa Clara, CA, USA).

The high-resolution electrospray mass spectrometric (HR-ESI-MS) detection was performed at the Center of Collective Use “Mass spectrometric investigations” SB RAS in positive mode within the 500–3000 *m*/*z* range on an electrospray ionization quadrupole time-of-flight (ESI-q-TOF) high-resolution mass spectrometer Maxis 4G (Bruker Daltonics, Bremen, Germany). The 1D and 2D NMR spectra of sample were obtained from CD_3_OD solution at room temperature on a Bruker Avance III 500 FT-spectrometer (Bruker BioSpin AG, Faellanden, Switzerland) with working frequencies 500.03, 125.73, 95.36, and 50.67 MHz for ^1^H, ^13^C, ^77^Se, and ^15^N, respectively. Due to a limited excitation width, we were unable to simultaneously acquire both signals of the ^77^Se nuclei. Two different experiments with a spectral width 100 ppm were carried out over 12 h. Experiments were performed using a 5.0 s relaxation delay and 3.3s acquisition times. The ^1^H and ^13^C NMR chemical shifts are reported in ppm of the δ scale and refer to the signal of the methyl group of the solvent (δ = 3.31 ppm for residual protons for the ^1^H- and 49.0 ppm for ^13^C-NMR spectra). The ^77^Se and ^15^N NMR chemical shifts refer to external standards of 1M SeO_2_ solution in D_2_O (δ(^77^Se) = 1282 ppm) and formamide (δ (^15^N) = 112.5 ppm). ^15^N NMR spectrum was obtained as a projection of 2-D ^1^H–^15^N-correlation. Assignment of the signals was carried out using 2D (HSQC, HMBC) NMR techniques.

The thermal properties (TGA) were studied on a Thermo Microbalance TG 209 F3 Tarsus (NETZSCH, Selb, Germany) from 25 to 850 °C at a heating rate of 10 °C·min^−1^ in He flow (30 mL·min^−1^). Powder X-ray diffraction (PXRD) patterns were collected on a Shimadzu XRD 7000S diffractometer (Shimadzu, Kyoto, Japan) (CuK_α_ radiation, graphite monochromator and Si as an external reference).

X-ray photoelectron spectroscopy (XPS) was performed on a FleXPS spectrometer equipped with a 1D-DLD detector system (Specs GmbH, Berlin, Germany) with monochromatic Al Kα excitation (1486.61 eV). The electron pass energy was 20 eV. The powder samples were pressed into double-sided adhesive Cu tape. Calibration of the binding energies was performed relative to an internal standard from the C 1s to 285.0 eV. Separation of the contributions from different atoms was carried out by a fitting of spectra on mixed Lorentzian−Gaussian symmetrical components.

### 3.4. Single-Crystal X-ray Diffraction Analysis (XRD)

Single-crystal X-ray diffraction data for **Mo_5_red** and **Mo_5_ox** were collected at 150 K on a Bruker Apex DUO diffractometer (Bruker Corporation, Billerica, MA, USA) fitted with graphite monochromatized MoKα radiation (λ = 0.71073 Å). Absorption corrections were made empirically using the SADABS program [54]. The structures were solved by the direct method and further refined by the full-matrix least-squares method using the SHELXTL program package [54]. All non-hydrogen atoms were refined anisotropically. Appendix A summarizes the crystallographic data, while CCDC 2219811–2219812 contain the supplementary crystallographic data for this paper. These data can be obtained free of charge from the Cambridge Crystallographic Data Centre via www.ccdc.cam.ac.uk/data_request/cif (accessed on 21 January 2023).

### 3.5. Cyclic Voltammetry

Cyclic voltammetry was carried out with Elins P-20X8 voltammetry analyzer (Electrochemical Instruments, Chernogolovka, Russia) using a three-electrode scheme with GC working, Pt auxiliary, and Ag/AgCl/3.5M KCl reference electrodes. Investigations were carried out for 5·10^–4^ M solution of corresponding cluster compound in 0.1 M Bu_4_NClO_4_ in acetonitrile under Ar atmosphere.

### 3.6. DFT Calculations

Density functional theory (DFT) calculations were carried out for [{Mo_5_(μ_3_-Se)_4_(μ_4_-Se)(μ-pz)_4_}(pzH)_5_]^2+^ cluster anion in the ADF2017 software package [55]. Geometric parameters were optimized with VWN + S12g dispersion-corrected density functional [56,57,58] and all-electron TZP basis set [59]. The calculated vibrational spectrum contained no imaginary frequencies. Single-point calculations of bonding energies and molecular orbitals with geometry from the VWN + S12g/TZP level of theory were carried out with a dispersion-corrected hybrid-density functional S12h and all-electron TZP basis set. The zero-order regular approximation (ZORA) was used in all calculations in this work to take into account the scalar relativistic effects [60,61,62]. All calculations were performed using the CH_3_CN environment effects, which were added with the Conductor-like Screening Model (COSMO) model [63].

The g-tensor was calculated using a spin-orbit coupled spin unrestricted relativistic ZORA method [60,61,62], and bond orders were computed using the Nalewajski–Mrozek spin unrestricted relativistic ZORA method [64,65,66] as implemented in the ADF2021 software package [55]. These calculations were performed with hybrid B3LYP functional [67] and all-electron basis sets TZP for all atoms [59]. A collinear approximation of the spin density was used in a spin-orbit coupled calculation. The structures of [{Mo_5_(μ_3_-Se)_4_(μ_4_-Se)(μ-pz)_4_}(pzH)_5_]^2+^ and [{Mo_5_(μ_3_-Se)_4_(μ_4_-Se)(μ-pz)_4_}(pzH)_5_]^+^ were taken from structural data and were not optimized, except for the hydrogen atoms of apical pyrazole ligand, whose positions were obtained from spin-unrestricted scalar ZORA [60] relativistic computation with BP86 [68,69] GGA functional and all-electron TZP basis sets [59].

### 3.7. EPR

The X-band continuous-wave EPR spectra were recorded at 77 and 300K with a Varian E-109 spectrometer. The frequency of the spectrometer was calibrated with a 2,2-diphenyl-1-picrylhydrazyl (DPPH) standard sample. The weighted portion of copper(II) sulfate pentahydrate (CuSO_4_·5H_2_O) was used to evaluate the concentration of paramagnetic species. The EPR spectra were simulated in the MATLAB program package with the EasySpin toolbox [70].

### 3.8. Magnetic Susceptibility

The magnetic properties of the samples studied were measured using the Faraday method in the temperature range of 80–300 K. The temperature stabilization was controlled using a Delta DTB9696 temperature controller. The voltage from a quartz torque microbalance was measured using high-precision Keysight 34465A digital voltmeter (Keysight Technologies, Santa Rosa, CA, USA). The magnetic field strength was 8.6 kOe. The powder samples (~30 mg) were placed in the open quartz ampoules and vacuumed at 0.01 Torr pressure. During the measurements, the samples were held in a rarefied helium atmosphere of 5 Torr pressure.

## 4. Conclusions

New and unusual representatives of the chalcogenide molybdenum cluster family were obtained and characterized in detail. The square-pyramidal cluster [{Mo_5_(μ_3_-Se)^i^_4_(μ_4_-Se)^i^(μ-pz)^i^_4_}(pzH)^t^_5_]Br synthesized from Mo_6_Br_12_ is built by both inner chalcogenide μ_3_-Se/μ_4_-Se and pyrazolate ligands and terminal pyrazole ligands. The complex is paramagnetic (15 VEC), and its properties have been studied by EPR and magnetic susceptibility experiments. Upon dissolution, the cluster oxidized by atmospheric oxygen to form [{Mo_5_(μ_3_-Se)^i^_4_(μ_4_-Se)^i^(μ-pz)^i^_4_}(pzH)^t^_5_]Br_2_. This compound is diamagnetic (14 VEC), and it exists in solution and in solids, which has been demonstrated in detail by a number of physicochemical methods of analysis (NMR spectroscopy, mass-spectrometry, XPS, etc.). Forms 1+ and 2+ reversibly transform into each other, as shown by cyclic voltammetry. Geometry, electronic structures, and energy levels of the new compounds was also analyzed by DFT calculations. The resulting complexes expand the chemistry of molybdenum chalcogenide clusters and are promising for use in catalysis, which will be studied in future works.

## Figures and Tables

**Figure 1 ijms-24-03440-f001:**
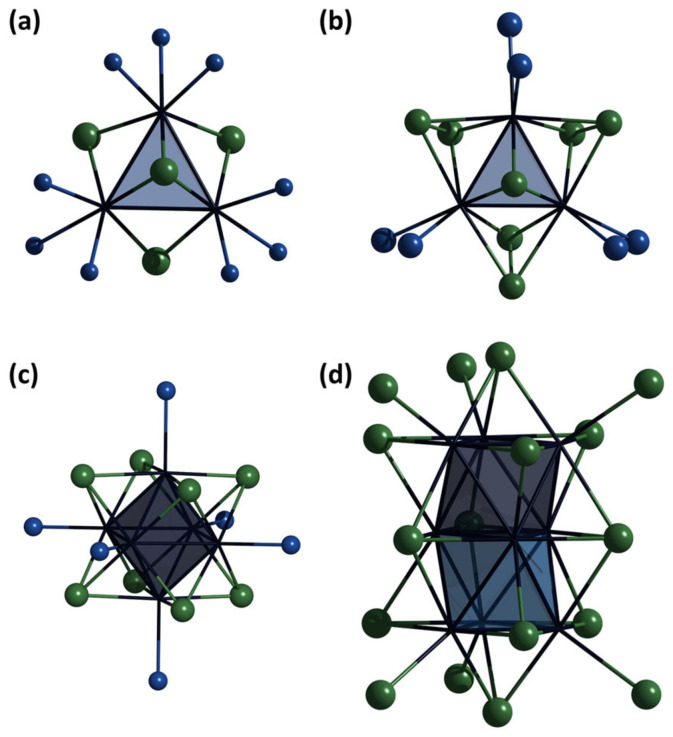
Structure of cluster complexes [Mo_3_Q_4_L_9_] (**a**), [Mo_3_Q_7_L_6_] (**b**), [Mo_6_Q_8_L_6_] (**c**), and [Mo_9_Se_11_] (**d**). Color code: Mo—dark blue, Mo_3_/Mo_6_—blue triangle/octahedron, Q—green, L—blue.

**Figure 2 ijms-24-03440-f002:**
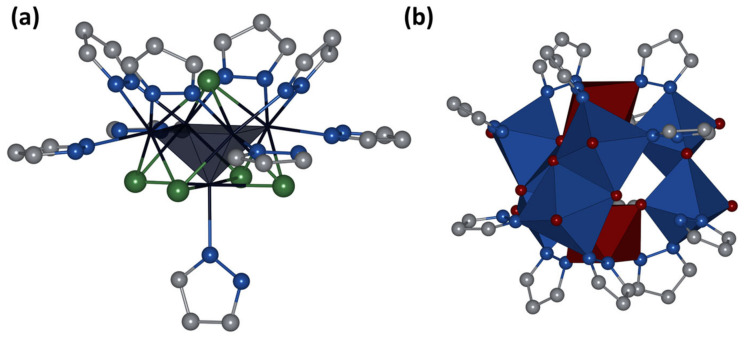
Crystal structure of cluster cation [{Mo_5_(μ_3_-Se)_4_(μ_4_-Se)(μ-pz)_4_}(pzH)_5_]^+^ in Mo_5_red (**a**) and hybrid POM from [31] (**b**). Hydrogen atoms are omitted for clarity. Color code: Mo—dark blue, Mo_5_—dark blue square pyramid, Se—green, C—gray, N—blue, Mo^V^_2_O_6_N_4_—blue edge-connected octahedrons, Mo^VI^O_3_N_3_—dark red octahedron.

**Figure 3 ijms-24-03440-f003:**
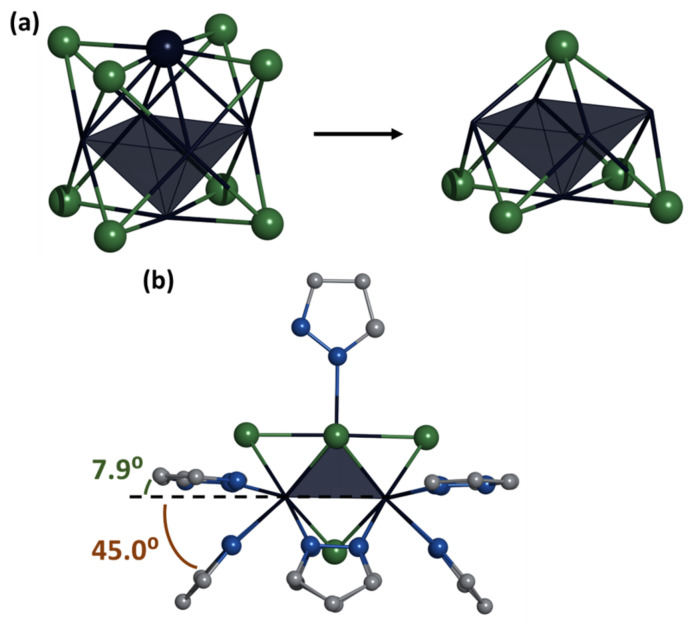
(**a**) Comparison of cluster cores {Mo_6_(μ_3_-Se)^i^_8_}^0^ and {Mo_5_(μ_3_-Se)^i^_4_(μ_4_-Se)^i^}^5+^. (**b**) Side view of cluster cation [{Mo_5_(μ_3_-Se)^i^_4_(μ_4_-Se)^i^(μ-pz)^i^_4_}(pzH)^t^_5_]^+^, demonstrating angles between inner or terminal ligands and base of pyramid. Hydrogen atoms are omitted for clarity. Color code: Mo—dark blue, Mo_5_—dark blue square pyramid, Se—green, C—gray, N—blue.

**Figure 4 ijms-24-03440-f004:**
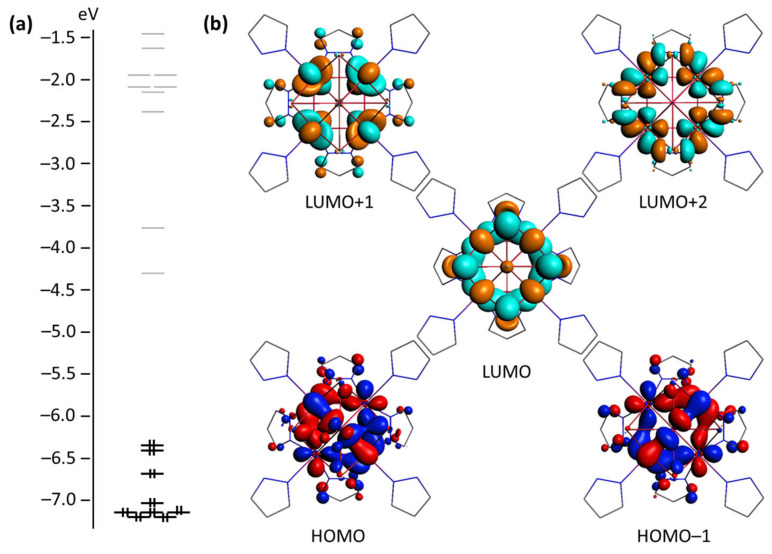
Energy levels diagram (**a**) and frontier molecular orbitals (**b**) of the [{Mo_5_(μ_3_-Se)_4_(μ_4_-Se)(μ-pz)_4_}(pzH)_5_]^2+^ cluster (top view representation along the C_4_ axis; pzH ligand coordinated to Mo^ap^ atom is hidden for clarity). Isosurface isovalue is 0.035 a.u.

**Figure 5 ijms-24-03440-f005:**
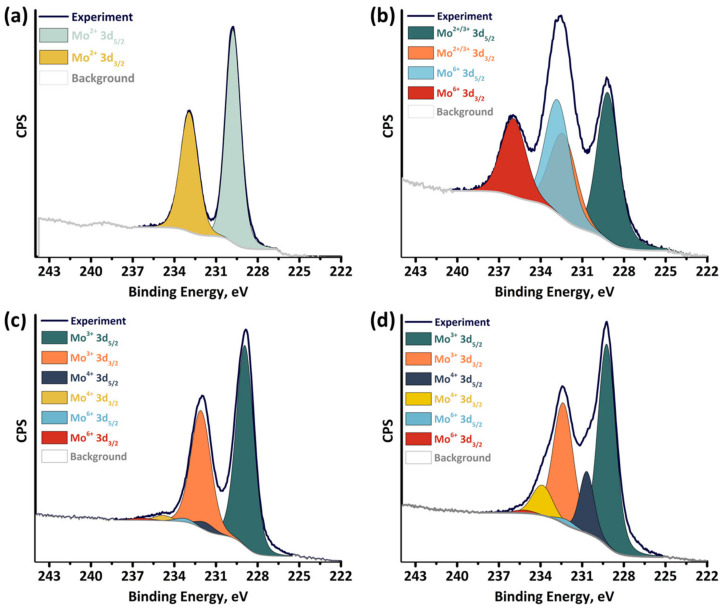
XPS spectra showing the Mo3d core levels in the Mo_6_Br_12_ (**a**), **Mo_6_** (**b**), **Mo_5_red** (**c**), and **Mo_5_ox** (**d**).

**Figure 6 ijms-24-03440-f006:**
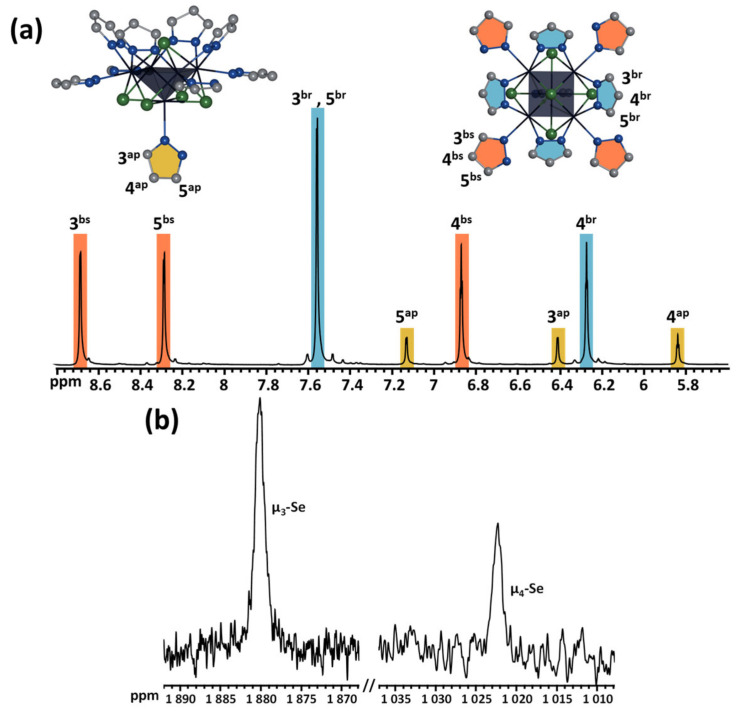
^1^H (**a**) and ^77^Se (**b**) NMR spectra of **Mo_5_ox** in CD_3_OD.

**Figure 7 ijms-24-03440-f007:**
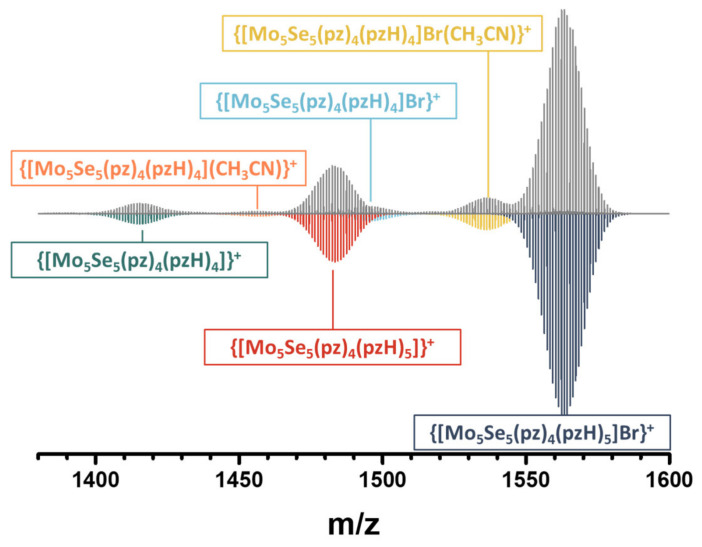
Fragment of HR-ESI-MS spectrum of solution of **Mo_5_ox** in acetonitrile (gray) and simulations of cluster forms (colored).

**Figure 8 ijms-24-03440-f008:**
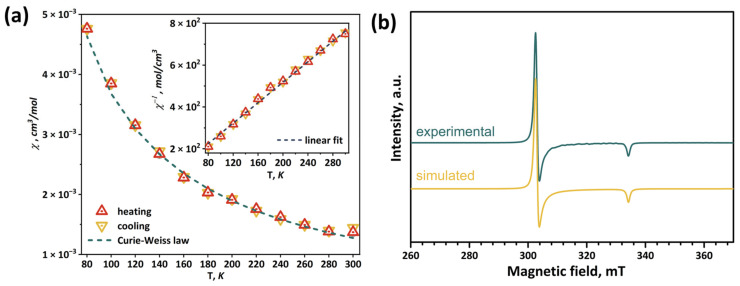
Temperature dependence of the magnetic susceptibility (**a**) and EPR spectrum measured at 77 K in comparison with simulated one (**b**) of **Mo5red**. Inset: linear fitting of inverse magnetic susceptibility.

**Figure 9 ijms-24-03440-f009:**
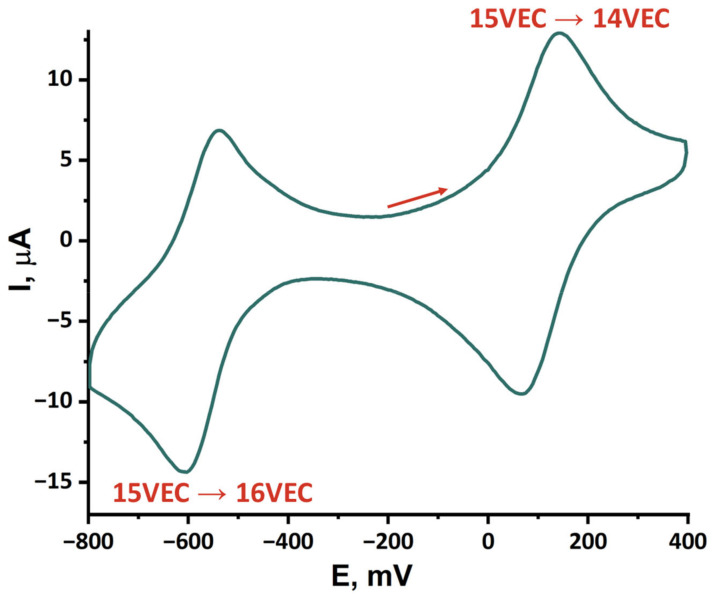
Cyclic voltammetry of the **Mo_5_red** (0.5 mM) in 0.1 M Bu_4_NClO_4_ acetonitrile solution, scan rate of 500 m vs. reference electrode, Ag/AgCl/3.5 M KCl.

**Table 2 ijms-24-03440-t002:** The bond order values resulting from the calculation for cluster ions [{Mo_5_(μ_3_-Se)_4_(μ_4_-Se)(μ-pz)_4_}(pzH)_5_]^+^ from **Mo_5_red** and [{Mo_5_(μ_3_-Se)_4_(μ_4_-Se)(μ-pz)_4_}(pzH)_5_]^2+^ from **Mo_5_ox**.

	Atom	Number	Mo-Mo	Mo-L	All
**Mo_5_red**	Mo^ap^	1	2.62	3.41	6.03
	Mo^bs^	4	1.73	3.95	5.68
	Total		9.55	19.20	28.75
**Mo_5_ox**	Mo^ap^	1	2.62	3.27	5.89
	Mo^bs^	4	1.62	4.17	5.79
	Total		9.11	19.95	29.06

**Table 3 ijms-24-03440-t003:** XPS Mo 3d_5/2–3/2_ binding energies (eV) in clusters obtained in comparison with literature data.

Compound	Mo 3d_5/2–3/2_ (%^(a)^), Assignment	Ref
**Mo_6_**	229.2–232.4 (54%), Mo^2+/3+^	This work
	232.8–236.0 (46%), Mo^6+^	
**Mo_5_red**	228.9–232.1 (95%), Mo^3+^	This work
	231.7–234.8 (4%), Mo^4+^	
	233.3–236.4 (1%), Mo^6+^	
**Mo_5_ox**	229.3–232.4 (78%), Mo^3+^	This work
	230.7–233.9 (20%), Mo^4+^	
	232.1–235.2 (2%), Mo^6+^	
Mo_6_Br_12_	229.5–232.5 (100%), Mo^2+^	[43]
Mo_6_Br_12_	229.8–232.9 (100%), Mo^2+^	This work
Mo_6_S_8_	228.5–231.7 (71%), Mo^2+^	[44]
	229.2–234.4 (24%), Mo^3+^	
	233.3–236.5 (5%), Mo^6+^	
[Mo_6_Se_8_(PEt_3_)_6_]	227.8–230.8, Mo^2+^ and Mo^3+^	[5]
K_2_[Mo_3_S_4_(Hnta)_3_]	230.1–233.3 (100%), Mo^4+^	[45]
(NH_4_)_2_[Mo_3_S_13_]	229.0–232.5 (100%), Mo^4+^	[46]
Mo_3_Se_13_	228.7–231.4, Mo^4+^	[47]
	232.1–235.0, Mo^6+^	

(a) The relative percentages of the different doublets are indicated.

**Table 4 ijms-24-03440-t004:** Main electrochemical potentials (Volt vs. Ag/AgCl) for chalcogenide clusters in 0.1 M Bu_4_NClO_4_ acetonitrile solution unless otherwise stated.

Compound	Process ^(a)^	E_a_	E_c_	E_1/2_	Ref
Mo_5_red	Mo^III^_5_ to Mo^III^_4_Mo^II^, qrev	−0.54	−0.60	−0.57	This work
	Mo^III^_5_ to Mo^III^_4_Mo^IV^, rev	0.14	0.07	0.11	
[Mo_3_Se_4_Br_3_(dmpe)_3_]PF_6_ ^(b)^	Mo^IV^_3_ to Mo^III^Mo^IV^_2_, rev	–	–	−0.54	[12]
	Mo^III^Mo^IV^_2_ to Mo^III^_2_Mo^IV^, rev	–	–	−0.87	
	Mo^IV^_3_ to Mo^IV^Mo^V^_2_, rev	–	–	1.10	
(NMe_4_)_3_[Mo_4_Se_4_(edta)_2_] ^(c)^	Mo^III^_3_Mo^IV^ to Mo^III^_4_, rev	−0.20	−0.28	−0.24	[52]
	Mo^III^_3_Mo^IV^ to Mo^III^_2_Mo^IV^_2_, rev	0.48	0.42	0.45	
[Mo_6_Se_8_(PEt_3_)_6_] ^(d)^	Mo^II^_2_Mo^III^_4_ to Mo^II^_3_Mo^III^_3_, rev	–	–	−0.96	[5]
	Mo^II^_2_Mo^III^_4_ to Mo^II^Mo^III^_5_, rev	–	–	0.31	
(Bu_4_N)_2_[Mo_5_Br_13_] ^(e)^	Mo^II^_4_Mo^III^ to Mo^II^_5_, rev	–	–	−0.33	[50]
	Mo^II^_4_Mo^III^ to Mo^II^_3_Mo^III^_2_, rev	–	–	0.41	

^(a)^ qrev = quasi-reversible, rev = reversible, irrev = irreversible; ^(b)^ dmpe = 1,2-bis(dimethylphosphino)ethane, CV were in recorded in 0.1 M Bu_4_NPF_6_ acetonitrile solution; ^(c)^ in 0.1 LiClO_4_ water solution, the potentials were recalculated from published data vs. SHE; ^(d)^ in Bu_4_NBF_4_ methylene chloride solution, the potentials were recalculated from published data vs. Fc/Fc^+^; ^(e)^ in 0.2M Bu_4_NPF_6_ methylene chloride solution.

## Data Availability

Crystal structure data can be obtained free of charge from The Cambridge Crystallographic Data Centre via www.ccdc.cam.ac.uk/data_request/cif (accessed on 21 January 2023) or are available on request from the corresponding author.

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
