# Peer review of "Unusual Square Pyramidal Chalcogenide Mo5 Cluster with Bridging Pyrazolate-Ligands"

_ijms, 2023, doi:10.3390/ijms24043440_

Round 1
Reviewer 1 Report
The MS introduces novel Mo5 cluster as the mixture of the reduced and oxidized forms. However, a manual separation of differently colored crystals can be hardly considered as a prepative technique. In this connection, it is required to provide the methods allowing to get pure samples of reduced and oxidized cluster forms.
The authors should explain the suitability of their results to the chosen section "Macromolecules". As for me they should look for another section.
Author Response
The MS introduces novel Mo5 cluster as the mixture of the reduced and oxidized forms. However, a manual separation of differently colored crystals can be hardly considered as a prepative technique. In this connection, it is required to provide the methods allowing to get pure samples of reduced and oxidized cluster forms.
Answer: During the synthesis in the melt of pyrazole, only the crystals of the reduced Mo5 form were obtained. Dissolution and separation from the reaction mixture yielded the oxidized form. Unfortunately, we haven't found an effective way to separate the reduced form except by manual separation. Obtaining the oxidized form is much easier and more preparative. Nevertheless, according to our experiments, the Mo5ox form can be reduced by common reducing agents (for example, hydrazine), which allows us to obtain Mo5red in solution. To clarify the synthesis and preparation of Mo5red and Mo5ox forms individually, additional discussion has been added to the manuscript.
The authors should explain the suitability of their results to the chosen section "Macromolecules". As for me they should look for another section.
Answer: Thank you for your comment. We have changed the section to Materials Science, which is more appropriate for our work.
Reviewer 2 Report
Report for the Manuscript Number IJMS-2209674
The manuscript by Savina et al. reports the syntheses and structural characterization, the spectroscopic, redox and magnetic properties, as well as the theoretical calculations of two novel selenide-bridged pentanuclear molybdenum complexes with pyrazole as bridging and terminal ligand. The results reported herein represent a further contribution to the well-known chemistry of chalcogenide metal clusters (see ref. [2]), which has been extensively studied by several groups since the 1960s up to nowadays because of their unique redox and catalytic activities or magnetic and electronic (conducting or superconducting) properties in order to obtain a new and evergreen class of advanced multifunctional molecular materials for nanoscience and nanotechnology.
In particular, the present work expands the previous work by some of the authors on the chemistry of chalcogenide molybdenum clusters (see refs. [7], [27], and [39]) by focusing on the use of pyrazole as N-donor ligand for the preparation of a novel pair of selenide-bridged pentanuclear molybdenum complexes from the parent hexanuclear precursor with the P-donor triphenylphosphine ligand, following a rare ligand exchange-triggered structural rearrangement of the metal core. In this respect, they have previously reported a related family of selenide- and telluride-bridged hexanuclear rhenium complexes with pyrazole and 3,5-dimethylpyrazole as terminal ligands (see ref. [29]), as well as sulphide- and selenide-bridged hexanuclear tungsten complexes with propionic acid-functionalized diphenylphosphine as terminal ligand featuring promising biomedical applications (see ref. [48]).
Overall, the synthetic and structural work is well done and satisfactorily rationalized, and the spectroscopic, redox and magnetic properties are interesting and correctly interpreted with the aid of DFT calculations. These two novel pyrazole-containing chalcogenide molybdenum clusters constitute thus a unique pair of electron-transfer (redox) isomers with an unusual square pyramidal Mo5 geometry. The majority of examples reported in the literature are triangular Mo3 and octahedral Mo6 clusters or Mon (n = 2, 4, 9, 12 and 15) with varying geometries (see refs. [4]-[11]). Besides, the presence of pyrazole as additional bridging coligand in the formation of the cluster core is remarkable from the structural and electronic viewpoint. As stated by the authors, the only examples of chalcogenide molybdenum clusters with pyrazole acting as additional bridging coligand are two oxo- or sulfide-bridged Mo8 and Mo4 complexes (see refs. [30] and [31]). Interestingly, the initial MoIII5 cluster show an interesting dual (capacitor-like) redox behavior, being reversible oxidized and reduced to afford the mixed-valence MoIII4MoIV and presumably MoIII4MoII species.
Therefore, I strongly recommend this paper for publication in International Journal of Molecular Sciences.However, it may be better considered for the Materials section of the journal, rather than for the Macromolecules one. Minor issues and comments that should/would be addressed by the authors are listed hereunder:
1. Besides the unusual square pyramidal geometry of the chalcogenide Mo5 clusters, the bridging mode of pyrazole should be also emphasized in the title.
2. For reasons of clarity and conciseness, the authors should consider the renaming of some of the section headings (v.g., 2.1. Synthesis; 2.5. NMR and HR-ESI-MS spectroscopy; 2.6. Magnetic properties and EPR spectroscopy), as well as adding and additional section including only the cyclic voltammetric results which should be separated from the magnetic ones (i.e., 2.7. Redox properties). Besides, the paragraph devoted to the preparation of the oxidized cluster, likely by air oxidation of the reduced one (lines 116-130), should be moved from the middle of the structural discussion (section 2.2) to the end of the synthetic one (section 2.1).
3. In the introduction section, the authors should be aware of the main role of pyrazole as bridging unit and magnetic coupler in dicopper(II) metallacyclic complexes, so-called pyrazolenophanes, for the preparation of functional magnetic compounds (see, for instance, the recent review by I. Castro et al. in Coord. Chem. Rev.2016, 315, 135-152 and refs. therein).
4. The terms “axial” and “equatorial” refer to an octahedral geometry. In this case, the authors should alternatively use “apical” and “basal” notation when referred to the square pyramidal geometry of their Mo5 clusters, both in the text and the tables of the manuscript.
5. The oxidation locus in this pair of redox isomers is indeed difficult to stablish, as stated by the authors, given the likely delocalized mixed-valence nature of the oxidized MoIII4MoIV cluster. However, the molecular orbital DFT calculations on the optimized molecular structures of the reduced and oxidized molybdenum clusters indicate a LUMO for the oxidized MoIII4MoIV cluster (or, alternatively, a HOMO for the reduced MoIII5 cluster) displaying bonding character among the basal Mo atoms (see Figures 4 and S28). This picture agrees with the unexpected and overall basal metal-ligand bond shortening in the experimental molecular structures upon reduction (see Table 1). The authors could briefly comment on this point in the theoretical calculations discussion (section 2.3). Otherwise, the representation of the HOMO for the reduced MoIII5 cluster in Figure S28 is a little confusing because of the use of a ball-and-stick side view representation. An alternative stick-only top view representation (along the apical molecular axis) could be more appropriate to see the shape of the HOMO containing the unpaired electron (so-called “magnetic orbital”). Alternatively, a representation of the calculated spin density distribution should be even more informative regarding the spin delocalization and polarization effects in the paramagnetic reduced MoIII5 cluster with a S = ½ ground state.
6. The moderate value of the Weiss (not Curie) temperature (q = -6 K) for the reduced MoIII5 cluster is somewhat surprising given the expected very weak magnitude of the long-distance intercluster antiferromagnetic interactions through the N-H···N hydrogen bonds and C-H···p stacking interactions between the terminal and solvated pyrazole ligands or the bromide counteranions (see Figures S1 and S2), as well as the absence of intracluster zero-field splitting effects for a doublet (S = ½) ground state. If possible, the authors should perform additional direct current (DC) magnetic measurements down to 2 K to verify this point. Otherwise, a representation of cT against T or, alternatively, 1/c against T is more useful than that of c against T (see Figure 8a) to appreciate the deviation of the magnetic behavior from the Curie law temperature (q = 0).
7. On the other hand, alternating current (AC) magnetic and pulsed EPR measurements at low temperature on the reduced S = ½ MoIII5 cluster could be planned by the authors to investigate its presumed slow magnetic relaxation and quantum coherence properties. In fact, the related mixed-valence magnetic polyoxometallates (POMs) are recently proposed as good candidates to quantum bits (qubits) and multiqubit-based quantum gates for nanotechnological applications in molecular spintronics and quantum information processing (see, for instance, the recent review by J. J. Baldoví et al. in Adv. Inorg. Chem. 2017, 69, 213-249 and refs. therein). Hence, a reversible magnetic electroswitching behavior occurs in this redox pair between the paramagnetic S = ½ MoIII5 and the diamagnetic S = 0 MoIII4MoIV states, which can be then proposed as a new example of magnetic electroswitch (see, for instance, R. Rabelo et al. Chem. Commun. 2020, 56, 1242-1245).
8. Finally, the authors claim the occurrence of two additional quasi-reversible oxidations at high positive potentials for the reduced MoIII5 cluster (line 350), but they should be better considered as almost irreversible oxidations from the shape of the corresponding redox waves in the cyclic voltammogram (Figure S30).

Author Response
The manuscript by Savina et al. reports the syntheses and structural characterization, the spectroscopic, redox and magnetic properties, as well as the theoretical calculations of two novel selenide-bridged pentanuclear molybdenum complexes with pyrazole as bridging and terminal ligand. The results reported herein represent a further contribution to the well-known chemistry of chalcogenide metal clusters (see ref. [2]), which has been extensively studied by several groups since the 1960s up to nowadays because of their unique redox and catalytic activities or magnetic and electronic (conducting or superconducting) properties in order to obtain a new and evergreen class of advanced multifunctional molecular materials for nanoscience and nanotechnology. In particular, the present work expands the previous work by some of the authors on the chemistry of chalcogenide molybdenum clusters (see refs. [7], [27], and [39]) by focusing on the use of pyrazole as N-donor ligand for the preparation of a novel pair of selenide-bridged pentanuclear molybdenum complexes from the parent hexanuclear precursor with the P-donor triphenylphosphine ligand, following a rare ligand exchange-triggered structural rearrangement of the metal core. In this respect, they have previously reported a related family of selenide- and telluride-bridged hexanuclear rhenium complexes with pyrazole and 3,5-dimethylpyrazole as terminal ligands (see ref. [29]), as well as sulphide- and selenide-bridged hexanuclear tungsten complexes with propionic acid-functionalized diphenylphosphine as terminal ligand featuring promising biomedical applications (see ref. [48]). Overall, the synthetic and structural work is well done and satisfactorily rationalized, and the spectroscopic, redox and magnetic properties are interesting and correctly interpreted with the aid of DFT calculations. These two novel pyrazole-containing chalcogenide molybdenum clusters constitute thus a unique pair of electron-transfer (redox) isomers with an unusual square pyramidal Mo5 geometry. The majority of examples reported in the literature are triangular Mo3 and octahedral Mo6 clusters or Mon (n = 2, 4, 9, 12 and 15) with varying geometries (see refs. [4]-[11]). Besides, the presence of pyrazole as additional bridging coligand in the formation of the cluster core is remarkable from the structural and electronic viewpoint. As stated by the authors, the only examples of chalcogenide molybdenum clusters with pyrazole acting as additional bridging coligand are two oxo- or sulfide-bridged Mo8 and Mo4 complexes (see refs. [30] and [31]). Interestingly, the initial MoIII5 cluster show an interesting dual (capacitor-like) redox behavior, being reversible oxidized and reduced to afford the mixed-valence MoIII4MoIV and presumably MoIII4MoII species.
Therefore, I strongly recommend this paper for publication in International Journal of Molecular Sciences. However, it may be better considered for the Materials section of the journal, rather than for the Macromolecules one. Minor issues and comments that should/would be addressed by the authors are listed hereunder.
Answer: Thank you for your positive feedback and relevant suggestions, which we address below.
- Besides the unusual square pyramidal geometry of the chalcogenide Mo5clusters, the bridging mode of pyrazole should be also emphasized in the title.
Answer: The title of the manuscript has been changed according to suggestion of a reviewer.
- For reasons of clarity and conciseness, the authors should consider the renaming of some of the section headings (v.g.,2.1. Synthesis; 2.5. NMR and HR-ESI-MS spectroscopy; 2.6. Magnetic properties and EPR spectroscopy), as well as adding and additional section including only the cyclic voltammetric results which should be separated from the magnetic ones (i.e., 2.7. Redox properties). Besides, the paragraph devoted to the preparation of the oxidized cluster, likely by air oxidation of the reduced one (lines 116-130), should be moved from the middle of the structural discussion (section 2.2) to the end of the synthetic one (section 2.1).
Answer: Section headings have been renamed. An additional section (Redox properties) has been added. The text of the manuscript has been reorganized.
- In the introduction section, the authors should be aware of the main role of pyrazole as bridging unit and magnetic coupler in dicopper(II) metallacyclic complexes, so-called pyrazolenophanes, for the preparation of functional magnetic compounds (see, for instance, the recent review by I. Castro et al.in Coord. Chem. Rev.2016, 315, 135-152 and refs. therein).
Answer: The work mentioned is extremely interesting and certainly deserves a mention in the Introduction section. Additional discussion added to the article.
- The terms “axial” and “equatorial” refer to an octahedral geometry. In this case, the authors should alternatively use “apical” and “basal” notation when referred to the square pyramidal geometry of their Mo5clusters, both in the text and the tables of the manuscript.
Answer: The required corrections have been done.
- The oxidation locus in this pair of redox isomers is indeed difficult to stablish, as stated by the authors, given the likely delocalized mixed-valence nature of the oxidized MoIII4MoIVcluster. However, the molecular orbital DFT calculations on the optimized molecular structures of the reduced and oxidized molybdenum clusters indicate a LUMO for the oxidized MoIII4MoIVcluster (or, alternatively, a HOMO for the reduced MoIII5 cluster) displaying bonding character among the basal Mo atoms (see Figures 4 and S28). This picture agrees with the unexpected and overall basal metal-ligand bond shortening in the experimental molecular structures upon reduction (see Table 1). The authors could briefly comment on this point in the theoretical calculations discussion (section 2.3). Otherwise, the representation of the HOMO for the reduced MoIII5 cluster in Figure S28 is a little confusing because of the use of a ball-and-stick side view representation. An alternative stick-only top view representation (along the apical molecular axis) could be more appropriate to see the shape of the HOMO containing the unpaired electron (so-called “magnetic orbital”). Alternatively, a representation of the calculated spin density distribution should be even more informative regarding the spin delocalization and polarization effects in the paramagnetic reduced MoIII5 cluster with a S = ½ ground state.
Answer: Indeed, the LUMO for the oxidized cluster have a strongly bonding character among the basal Mo atoms, which agrees well with observed changes in geometry of the metal core upon reduction. The corresponding comment was added in the section 2.3. Additionally, Figures 4 and S28 were modified in such a way that molecular orbitals (and spin density distribution in Figure S28) are shown in stick-only top view representation. Also, as you requested, we have added the stick-only spin density plots along the [100] and [001] axes (Figure S29).
- The moderate value of the Weiss (not Curie) temperature (q = -6 K) for the reduced MoIII5cluster is somewhat surprising given the expected very weak magnitude of the long-distance intercluster antiferromagnetic interactions through the N-H···N hydrogen bonds and C-H···p stacking interactions between the terminal and solvated pyrazole ligands or the bromide counteranions (see Figures S1 and S2), as well as the absence of intracluster zero-field splitting effects for a doublet (S= ½) ground state. If possible, the authors should perform additional direct current (DC) magnetic measurements down to 2 K to verify this point. Otherwise, a representation of cT against T or, alternatively, 1/c against T is more useful than that of c against T (see Figure 8a) to appreciate the deviation of the magnetic behavior from the Curie law temperature (q = 0).
Answer: Thank you for your note. Unfortunately, we do not have the technical ability to perform the DC magnetic measurements in the low temperature region of 2 K. However, the linear fit of the 1/X(T) dependence indicates that the magnetic behavior of the sample under study obeys the Curie-Weiss law. This figure has been added to the inset of Figure 8a.
- On the other hand, alternating current (AC) magnetic and pulsed EPR measurements at low temperature on the reduced S= ½ MoIII5cluster could be planned by the authors to investigate its presumed slow magnetic relaxation and quantum coherence properties. In fact, the related mixed-valence magnetic polyoxometallates (POMs) are recently proposed as good candidates to quantum bits (qubits) and multiqubit-based quantum gates for nanotechnological applications in molecular spintronics and quantum information processing (see, for instance, the recent review by J. J. Baldoví et al. in Adv. Inorg. Chem. 2017, 69, 213-249 and refs. therein). Hence, a reversible magnetic electroswitching behavior occurs in this redox pair between the paramagnetic S = ½ MoIII5 and the diamagnetic S = 0 MoIII4MoIV states, which can be then proposed as a new example of magnetic electroswitch (see, for instance, R. Rabelo et al. Chem. Commun. 2020, 56, 12242-12245).
Answer: Thank you for the interesting advice! We do not believe it is necessary to conduct such studies for this publication. Nevertheless, we will certainly perform this study for our compounds in future work.
- Finally, the authors claim the occurrence of two additional quasi-reversible oxidations at high positive potentials for the reduced MoIII5cluster (line 350), but they should be better considered as almost irreversible oxidations from the shape of the corresponding redox waves in the cyclic voltammogram (Figure S30).
Answer: The correction has been made.